# The prognostic value of the direct bilirubin to albumin ratio in critically ill patients with cirrhosis: Insights from MIMIC-IV database

XingYi Yang[1], GuangDong Wang[2], Zhang Min[1], LiHong Lv[1], Ji Yang[1]*

1 Department of Gastroenterology Disease, XianJu People's Hospital, Zhejiang Southeast Campus of Zhejiang Provincial People's Hospital, Affiliated Xianju's Hospital, Hangzhou Medical College, Xianju, Zhejiang, China, 2 Department of Respiratory and Critical Care Medicine, First Affiliated Hospital of Xi'an Jiao tong University, Xi an, Shanxi, China

* yangji742903927@qq.com

## Abstract

### Background

Patients with severe cirrhosis are at a higher risk of mortality. This study aimed to investigate the association between the direct bilirubin-to-albumin ratio (DBAR) and 28-day mortality in critically ill cirrhotic patients using data from the publicly available MIMIC-IV database.

### Methods

This study explores DBAR's relationship with 28-day mortality in severe cirrhosis patients. We first conducted univariate and multivariate analyses to identify independent risk factors. Then, we used Kaplan-Meier (KM) survival analysis to assess DBAR's link with survival time and created KM curves. DBAR's predictive accuracy was evaluated using Receiver Operating Characteristic (ROC) analysis, and the relationship was examined using restricted cubic spline modeling and subgroup analyses.

### Result

This study enrolled 509 cirrhotic patients with in-hospital and ICU mortality rates of 22.3% and 14.3%, respectively. Univariate and multivariate analyses revealed a significant association between DBAR and 28-day mortality risk, with a hazard ratio of 1.16 (95% CI: 1.10–1.24, p < 0.001), confirming DBAR as an independent risk factor for short-term prognosis. DBAR demonstrated good predictive accuracy for 28-day mortality (AUC = 0.702, 95% CI: 0.650–0.753). Patients were divided into low-risk (DBAR < 4) and high-risk (DBAR ≥ 4) groups, with the high-risk group showing a hazard ratio of 3.05 (95% CI 1.87–4.97, p < 0.001) after multivariate adjustment. Restricted cubic spline (RCS) analysis identified a nonlinear relationship between

**Data availability statement:** All relevant data are within the manuscript and its Supporting Information files.

**Funding:** The author(s) received no specific funding for this work.

**Competing interests:** The authors have declared that no competing interests exist.

DBAR and 28-day prognosis (p-Nonlinear = 0.022, p < 0.001). Subgroup analysis showed no interaction between DBAR and most subgroups.

## Conclusion

The DBAR scoring system offers an efficient and user-friendly approach for assessing prognosis in critically ill cirrhotic patients.

---

## 1 Introduction

Cirrhosis, the terminal pathological manifestation of various chronic liver diseases, is characterized by persistent hepatocellular necrosis, excessive fibrous tissue deposition, and nodular parenchymal regeneration within the hepatic architecture [1,2]. These progressive structural alterations fundamentally disrupt the liver's lobular organization and vascular configuration, precipitating severe clinical consequences, particularly portal hypertension and hepatic decompensation, which significantly elevate both morbidity and mortality risks [3,4]. The admission of cirrhotic patients to intensive care units (ICUs) represents a critical clinical turning point, universally associated with poor prognosis [5,6]. This vulnerable population demonstrates increased predisposition to life-threatening complications, notably hepatic encephalopathy and septic events, which substantially complicate therapeutic interventions and markedly escalate mortality rates [7,8].

In the prognostic assessment of cirrhosis, both the Child-Pugh classification system [9] and the Model for End-Stage Liver Disease (MELD) score [10], despite their widespread clinical application, demonstrate significant limitations. The Child-Pugh system, in particular, is constrained by its dependence on subjective clinical parameters, especially in the assessment of hepatic encephalopathy, which reduces its reliability and consistency. This underscores an urgent need for the development of more precise, objective, and minimally invasive prognostic markers to improve predictive accuracy and clinical decision-making in cirrhotic patients.

Total bilirubin and albumin serve as critical biochemical markers in the prognostic evaluation of cirrhosis. Elevated bilirubin levels correlate with cholestatic progression and hepatocyte injury [11], whereas decreased albumin levels signify impaired hepatic synthetic function [12]. The bilirubin-albumin score has established prognostic utility in hepatic encephalopathy and hepatocellular carcinoma [13–15], with the indirect bilirubin/albumin ratio particularly demonstrating predictive accuracy for hepatic encephalopathy development [16]. Emerging evidence suggests that direct bilirubin exhibits superior prognostic value compared to total bilirubin levels in cirrhotic patients [17], highlighting the potential significance of the DBAR as a novel prognostic indicator. The link between DBAR and mortality in critically ill individuals with cirrhosis admitted to the ICU has not yet been thoroughly examined. To fill this research void, we performed a retrospective study utilizing clinical records from the MIMIC-IV v2.2 database [18], focusing on cirrhotic individuals. This study seeks to assess whether DBAR levels are connected to short-term mortality from all causes in this patient population.

## 2 Materials and methods

### 2.1 Database introduction

We utilized the MIMIC-IV database (version 2.2), a publicly accessible critical care dataset curated by the MIT Laboratory for Computational Physiology. The database contains detailed information on more than 70,000 ICU admissions at Beth Israel Deaconess Medical Center from 2012 to 2019. Ethical approval was granted by the Institutional Review Board of Beth Israel Deaconess Medical Center (protocol 2001P-001699/14). As all data were fully de-identified, further institutional approval and individual informed consent were waived.

### 2.2 Inclusion and exclusion criteria

In this retrospective cohort study, patients diagnosed with cirrhosis were included. The diagnosis of cirrhosis was determined by International Classification of Diseases codes. The study included 3683 patients with cirrhosis. Patients were excluded if they were under 18 years old, with HIV, spent less than 24 hours in the ICU, lacked direct bilirubin and albumin data within 24 hours of admission, or had previous ICU admissions. A total of 509 patients were ultimately included in the study. (Further details are delineated in Fig 1)

### 2.3 Data collection and monitoring

Data from the database were extracted, encompassing demographic information, initial clinical vital signs, laboratory results, comorbidities, and treatment outcomes, with a specific focus on the first 24 hours post-ICU admission. Key vital signs monitored included blood pressure, heart rate, respiratory rate, and body temperature. Comprehensive laboratory

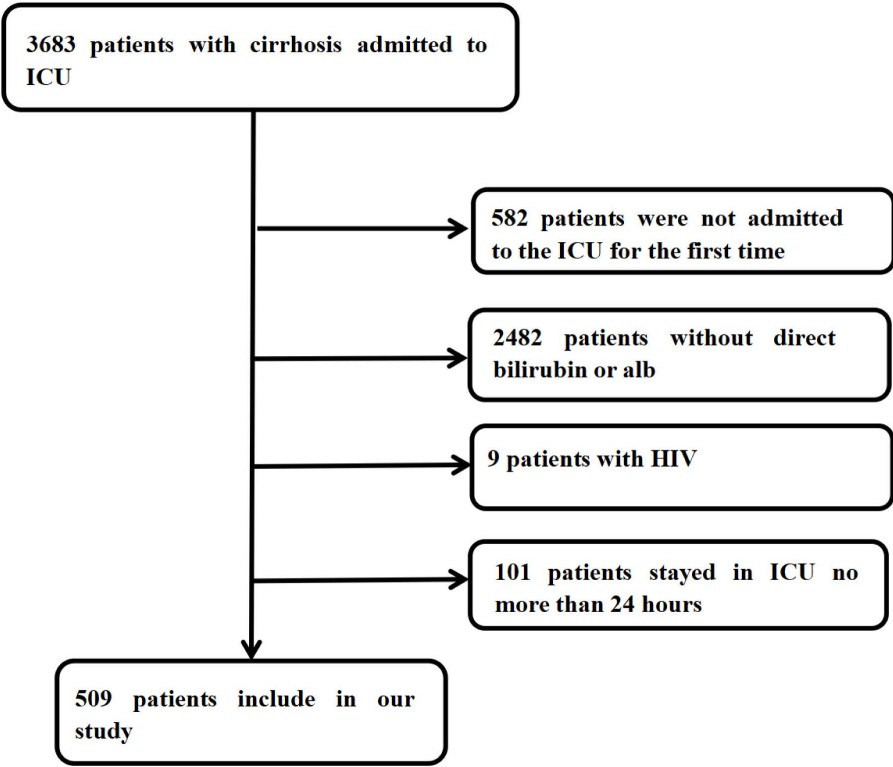

**Fig 1. Flow diagram of study participants.**

parameters were recorded, including blood glucose, white blood cell count (WBC), international normalized ratio (INR), platelet count, sodium, potassium, blood urea nitrogen (BUN), aminotransferase (ALT), and aspartate aminotransferase (AST), creatinine, direct bilirubin, albumin, and complications. Significant attention was devoted to the collection of data pertaining to complications associated with cirrhosis, including acute kidney injury (AKI), ascites, variceal hemorrhage, hepatorenal syndrome (HRS), and spontaneous bacterial peritonitis (SBP), alongside prevalent therapeutic interventions such as continuous renal replacement therapy (CRRT) and vasopressor administration. The DBAR was selected as the primary study variable, calculated as the average of values recorded within the initial 24 hours of ICU admission for all patients. The study cohort comprised individuals who met established diagnostic criteria for cirrhosis and were followed for a minimum of 28 days to evaluate short-term outcomes. Table 1 outlines the complete list of extracted variables in detail. The primary endpoint was defined as all-cause mortality at 28 days post-admission.

### 2.4 Statistical analyses

Continuous variables were reported as mean±SD or median (IQR), and categorical variables as counts (%). Baseline features were compared using Student's t test, ANOVA, chi-square, or Fisher's exact test. Variables with >20% missing values were excluded, while those with <20% were imputed using random forest. Univariate Cox regression analysis was conducted to identify significant variables, which were then included in the multivariate Cox regression model based on clinical relevance and statistical significance. The DBAR cut-off (DBAR<4 vs. ≥4) was determined using X-tile software (version 3.6.1, Yale University), with adjusted covariates to assess survival differences. KM curves and log-rank tests were used to evaluate survival probabilities. ROC analysis assessed the predictive performance of DBAR, direct bilirubin, albumin, and MELD scores, reporting sensitivity, specificity, and Area Under the Curve (AUC). In order to thoroughly evaluate the impact of various factors on the prediction of cirrhosis outcomes, we utilized the Boruta Feature Selection method, a highly regarded approach in the field of machine learning, to pinpoint the most critical predictive variables. RCS analysis explored potential nonlinear associations between DBAR and prognosis. Subgroup analyses examined the impact of covariates (age, sex, race, sepsis, SBP, ascites, HRS, vasoactive treatment) on the DBAR-mortality relationship. Analyses were performed using R (version 4.2.2, R Foundation).

## 3 Results

### 3.1 Baseline clinical characteristics

A total of 509 patients were included in this study, with a mean age of 59 years and a predominance of male (69.7%) and Caucasian (65.6%) individuals. Comparative analysis between the mortality and survival groups demonstrated that patients in the mortality group exhibited significant elevations in multiple clinical parameters, including heart rate, respiratory rate, systolic blood pressure, anion gap, BUN, potassium, WBC, direct bilirubin, creatinine and INR. Furthermore, the mortality group had substantially higher MELD scores (28.0 vs. 18.0). Additionally, the mortality group showed a higher prevalence of HRS (26.1% vs. 12%), ascites (69.4% vs. 52.3%), SBP (20.9% vs. 10.7%), AKI (92.5% vs. 82.1%), CRRT (26.9% vs. 10.7%), and vasopressor use (71.6% vs. 38.4%). Stratification based on DBAR revealed a stark contrast in survival rates between the two groups (58.4% vs. 21.6%). However, no statistically significant differences were observed in baseline demographics or clinical features, including age, diabetes, sepsis, and ventilator use (P≥0.05). The baseline characteristics of patients in both groups are detailed in Table 1.

### 3.2 Both univariate and multivariate Cox regression analyses were performed to identify clinical predictors of mortality during the 28-day follow-up

In the univariate Cox proportional hazards analysis, non-surviving patients exhibited significantly higher levels of age, DBAR, BUN, potassium, creatinine, lactate, WBC, INR, SBP, as well as increased utilization of CRRT and vasoactive

**Table 1. Patient demographics and baseline characteristics.**

| Variables | All patients (n = 509) | Survivors (n = 375) | Non-survivors (n = 134) | p-value |
|---|---|---|---|---|
| **Demographic** | | | | |
| Age (years) | 59 (52, 66) | 58 (51, 65) | 60 (52, 69) | 0.072 |
| Gender, n (%) | | | | 0.102 |
| Male | 355 (69.7%) | 269 (71.7%) | 86 (64.2%) | |
| Female | 154 (30.3%) | 106 (28.3%) | 48 (35.8%) | |
| Race, n (%) | | | | 0.067 |
| White | 334 (65.6%) | 254 (67.7%) | 80 (59.7%) | |
| Other | 175 (34.4%) | 121 (32.3%) | 54 (40.3%) | |
| **Vital Signs** | | | | |
| Heart rate (beats/minute) | 91 (79, 103) | 90 (79, 102) | 94 (81, 109) | 0.019 |
| Systolic blood pressure (mmHg) | 117 (102, 133) | 120 (106, 136) | 111 (94, 123) | <0.001 |
| Diastolic Blood Pressure (mmHg) | 62 (53, 71) | 62 (55, 71) | 61 (51, 69) | 0.124 |
| Respiratory rate (beats/min) | 18.0 (15.0, 22.0) | 17.0 (15.0, 20.0) | 19.5 (16.0, 25.0) | <0.001 |
| Temperature (°C) | 36.83(36.50, 37.20) | 36.89 (36.56, 37.28) | 36.61(36.22, 36.94) | <0.001 |
| **Laboratory Indicators** | | | | |
| Bicarbonate (m Eq/l) | 21.0 (18.0, 25.0) | 22.0 (19.0, 25.0) | 19.5 (16.0, 23.0) | <0.001 |
| Anion Gap (m Eq/l) | 16.0 (13.0, 19.0) | 16.0 (13.0, 19.0) | 17.0 (15.0, 21.0) | <0.001 |
| Ca (mg/dl) | 8.30 (7.70, 9.00) | 8.30 (7.80, 9.10) | 8.20 (7.50, 8.90) | 0.019 |
| BUN (mg/dl) | 23 (15, 44) | 20 (14, 34) | 39 (22, 65) | <0.001 |
| Potassium (m Eq/l) | 4.10 (3.70, 4.70) | 4.10 (3.70, 4.70) | 4.30 (3.70, 5.10) | 0.031 |
| Sodium (m Eq/l) | 138 (133, 141) | 138 (135, 141) | 134 (129, 139) | <0.001 |
| Glucose (mg/dl) | 141 (105, 198) | 156 (113, 214) | 114 (92, 144) | <0.001 |
| Creatinine (mg/dl) | 1.10 (0.80, 1.90) | 1.00 (0.75, 1.60) | 1.65 (1.00, 2.70) | <0.001 |
| Direct bilirubin (mg/dl) | 2.5 (1.2, 5.4) | 2.1 (1.1, 4.3) | 4.4 (2.2, 11.8) | <0.001 |
| Fibrinogen (mg/dl) | 181 (141, 228) | 184 (147, 228) | 169 (123, 231) | 0.046 |
| Lactate | 2.40 (1.70, 3.50) | 2.17 (1.50, 2.93) | 3.50 (2.50, 4.50) | <0.001 |
| ALT(IU/L) | 60 (27, 298) | 107 (28, 408) | 43 (27, 81) | <0.001 |
| ALP(IU/L) | 94 (66, 149) | 87 (64, 134) | 114 (82, 183) | <0.001 |
| AST(IU/L) | 141 (60, 579) | 205 (63, 731) | 100 (59, 189) | <0.001 |
| WBC (10^9/L) | 9(5, 13) | 8 (5, 12) | 11 (8, 19) | <0.001 |
| Platelet(10^9/L) | 94 (60, 138) | 93 (61, 131) | 105 (56, 153) | 0.613 |
| Hemoglobin(10^9/L) | 9.50 (8.15, 11.30) | 9.60 (8.30, 11.40) | 9.20 (7.70, 10.70) | 0.060 |
| INR | 1.70 (1.40, 2.25) | 1.60 (1.30, 2.00) | 2.10 (1.80, 2.80) | <0.001 |
| PT(s) | 19 (16, 24) | 18 (15, 22) | 23 (19, 29) | <0.001 |
| APTT(s) | 39 (33, 50) | 38 (32, 47) | 42 (36, 57) | <0.001 |
| Albumin(mg/dl) | 2.90 (2.50, 3.20) | 2.90 (2.50, 3.20) | 2.70 (2.40, 3.20) | 0.093 |
| Chloride (m Eq/l) | 103 (98, 106) | 104 (100, 106) | 100 (94, 105) | <0.001 |
| DBAR | 0.90 (0.45, 2.13) | 0.75 (0.38, 1.56) | 1.63 (0.83, 4.29) | <0.001 |
| MELD | 20.0 (13.0, 28.0) | 18.0 (12.0, 25.0) | 28.0 (21.0, 34.0) | <0.001 |
| **Comorbidities** | | | | |
| Diabetes, n (%) | 127 (25.0%) | 101 (26.9%) | 26 (19.4%) | 0.063 |
| Renal disease, n (%) | 74 (14.5%) | 48 (12.8%) | 26 (19.4%) | 0.063 |
| AKI, n (%) | 432 (84.9%) | 308 (82.1%) | 124 (92.5%) | 0.046 |
| Sepsis, n (%) | 411 (80.7%) | 296 (78.9%) | 115 (85.8%) | 0.083 |
| SIRS | 457 (89.8%) | 333 (88.8%) | 124 (92.5%) | 0.220 |
| Hepatorenal syndrome, n (%) | 80 (15.7%) | 45 (12.0%) | 35 (26.1%) | <0.001 |

*(Continued)*

**Table 1.** (Continued)

| Variables | All patients (n = 509) | Survivors (n = 375) | Non-survivors (n = 134) | p-value |
|---|---|---|---|---|
| Variceal bleeding, n (%) | 17 (3.3%) | 14 (3.7%) | 3 (2.2%) | 0.578 |
| Ascites, n (%) | 289 (56.8%) | 196 (52.3%) | 93 (69.4%) | <0.001 |
| Spontaneous bacterial peritonitis, n (%) | 68 (13.4%) | 40 (10.7%) | 28 (20.9%) | 0.003 |
| **Treatment** | | | | |
| Vasoactive, n (%) | 240 (47.2%) | 144 (38.4%) | 96 (71.6%) | <0.001 |
| Ventilator, n (%) | 417 (81.9%) | 310 (82.7%) | 107 (79.9%) | 0.467 |
| CRRT, n (%) | 76 (14.9%) | 40 (10.7%) | 36 (26.9%) | <0.001 |
| **Group** | | | | |
| DBAR (<4), n (%) | 444 (87.2%) | 348 (92.8%) | 96 (71.6%) | <0.001 |
| DBAR (≥4), n (%) | 65 (12.8%) | 27 (7.2%) | 38 (28.4%) | |

Ca: calcium; BUN: Blood Urea Nitrogen; AST: aspartate aminotransferase; ALT: alanine aminotransferase; WBC: white blood cell; INR: international normalized ratio; PT: prothrombin time; APTT: Activated Partial Thromboplastin Time; ALP: Alkaline Phosphatase; LDH: Lactate Dehydrogenase; SIRS: Systemic Inflammatory Response Syndrome; MELD: Model for End-Stage Liver Disease; CRRT: continuous renal replacement therapy; AKI: acute kidney injury; DBAR: Direct Bilirubin-to-Albumin Ratio.

agents. Additionally, a higher prevalence of SBP, hepatorenal syndrome, and ascites was observed in this group. Conversely, lower levels of sodium, chloride, ALT, and AST were noted in non-survivors.

In the multivariate Cox proportional hazards model, several variables were independently associated with 28-day mortality, including DBAR (HR = 1.16, 95% CI 1.10–1.24; p < 0.001), Age (HR = 1.03, 95% CI 1.02–1.05; p < 0.001), BUN (HR = 1.01, 95% CI 1.00–1.01; p = 0.006), Lactate (HR = 1.21, 95% CI 1.15–1.27; p < 0.001), ALT (HR = 1.00, 95% CI 1.00–1.00; p < 0.001), INR (HR = 1.17, 95% CI 1.01–1.37; p = 0.041), and vasoactive agent use (HR = 2.56, 95% CI 1.72–3.81; p < 0.001). Detailed results are presented in Table 2.

### 3.3 Relationship between DBAR and 28-day mortality

In the multivariate Cox regression analysis, variables were selected for inclusion based on their statistical significance in univariate analysis or their established clinical relevance to ensure accurate model adjustment. In the initial model (Model 1), the HR for the association between DBAR and 28-day mortality was 1.21 (95% CI 1.15–1.26; p < 0.001). Model 2, which incorporated additional clinically pertinent covariates, demonstrated a slightly attenuated HR for DBAR of 1.15 (95% CI 1.09–1.21; p < 0.001). Extending Model 2, Model 3 further reinforced the independent prognostic value of DBAR, with an HR of 1.17 (95% CI 1.09–1.26; p < 0.001). These findings robustly establish DBAR as a significant and independent predictor of 28-day mortality among critically ill cirrhotic patients with cirrhosis.

To investigate the differential impact of DBAR levels, we utilized X-tile software to stratify patients according to 28-day mortality. The cohort was categorized into two distinct groups: a high-risk group (DBAR ≥ 4) and a low-risk group (DBAR < 4). In Model 1, which adjusted for age, sex, and race, the high-risk group exhibited a significantly elevated mortality rate, with a HR of 4.82 (95% CI 3.22–7.23; p < 0.001). Further adjustments in Model 2 and Model 3 consistently affirmed the heightened mortality risk in the high-risk group, highlighting a robust and independent association between DBAR ≥ 4 and mortality. Specifically, Model 2 yielded an HR of 3.22 (95% CI 2.10–5.24; p < 0.001), and Model 3 produced an HR of 3.05 (95% CI 1.87–4.97; p < 0.001). Comprehensive results are summarized in Table 3.

### 3.4 Analysis of Kaplan-Meier and ROC curves

A KM survival curve analysis demonstrated a significantly higher mortality rate at the 28-day follow-up for patients with a DBAR ≥ 4 compared to those with a DBAR < 4 (56.9% vs. 18.4%, respectively; Fig 2). The AUC for DBAR in predicting

**Table 2. Cox regression analyses of clinical parameters associated with 28-day mortality.**

| Variables | Univariate analysis | | | Multivariate analysis | | |
|---|---|---|---|---|---|---|
| | HR | 95% CI | p-value | HR | 95% CI | p-value |
| Age(year) | 1.02 | 1.00, 1.04 | 0.016 | 1.03 | 1.02, 1.05 | <0.001 |
| Gender, n (%) | | | | | | |
| Male | | | | | | |
| Female | 1.34 | 0.94, 1.91 | 0.104 | | | |
| Race, n (%) | | | | | | |
| Other | – | – | – | | | |
| White | 0.73 | 0.52,1.03 | 0.072 | | | |
| Respiratory Rate(beats/min) | 1.06 | 1.04, 1.09 | <0.001 | | | |
| Heart Rate(beats/min) | 1.01 | 1.00, 1.02 | 0.003 | | | |
| Temperature (°C) | 0.65 | 0.57, 0.74 | <0.001 | | | |
| DBAR | 1.18 | 1.13, 1.24 | <0.001 | 1.16 | 1.10, 1.24 | <0.001 |
| BUN (mg/dl) | 1.02 | 1.01, 1.02 | <0.001 | 1.01 | 1.00, 1.01 | 0.006 |
| Bicarbonate (m Eq/l) | 0.90 | 0.86, 0.93 | <0.001 | | | |
| Potassium (m Eq/l) | 1.35 | 1.13, 1.60 | <0.001 | | | |
| Sodium (m Eq/l) | 0.95 | 0.93, 0.97 | <0.001 | | | |
| Creatinine (mg/dl) | 1.16 | 1.08, 1.24 | <0.001 | | | |
| Fibrinogen(mg/dl) | 1.00 | 1.00, 1.00 | 0.588 | | | |
| Lactate (mmol/L) | 1.21 | 1.17, 1.26 | <0.001 | 1.21 | 1.15, 1.27 | <0.001 |
| ALT(IU/L) | 1.00 | 1.00, 1.00 | <0.001 | 1.00 | 1.00, 1.00 | <0.001 |
| AST(IU/L) | 1.00 | 1.00, 1.00 | 0.012 | | | |
| WBC (10^9/L) | 1.06 | 1.04, 1.07 | <0.001 | | | |
| Platelet(10^9/L) | 1.00 | 1.00, 1.00 | 0.454 | | | |
| Hemoglobin(10^9/L) | 0.94 | 0.87, 1.02 | 0.135 | | | |
| INR | 1.67 | 1.47, 1.90 | <0.001 | 1.17 | 1.01, 1.37 | 0.041 |
| Chloride (m Eq/l) | 0.97 | 0.95, 0.99 | <0.001 | | | |
| AKI, n (%) | 2.47 | 1.29, 4.70 | 0.006 | | | |
| Sepsis, n (%) | 1.55 | 0.96, 2.53 | 0.075 | | | |
| Spontaneous bacterial peritonitis, n (%) | 1.80 | 1.19, 2.73 | 0.006 | | | |
| Hepatorenal syndrome, n (%) | 2.12 | 1.44, 3.12 | <0.001 | | | |
| SIRS, n (%) | 1.52 | 0.80, 2.89 | 0.205 | | | |
| Variceal bleeding, n (%) | 0.60 | 0.19, 1.88 | 0.377 | | | |
| Ascites, n (%) | 1.82 | 1.26, 2.62 | 0.001 | | | |
| Vasoactive, n (%) | 3.42 | 2.35, 4.98 | <0.001 | 2.56 | 1.72, 3.81 | <0.001 |
| Ventilator, n (%) | 0.87 | 0.57, 1.32 | 0.502 | | | |
| CRRT, n (%) | 2.51 | 1.71, 3.68 | <0.001 | | | |

Ca, calcium; BUN, Blood Urea Nitrogen; AST, aspartate aminotransferase; ALT, alanine aminotransferase; WBC, white blood cell; INR, international normalized ratio; LDH: Lactate Dehydrogenase; SIRS: Systemic Inflammatory Response Syndrome; CRRT, continuous renal replacement therapy; AKI, acute kidney injury; DBAR: Direct Bilirubin-to-Albumin Ratio; HR: Hazard Ratio; 95 CI: 95% Confidence Interval.

short-term mortality among this patient population was 0.702 (95% CI: 0.650–0.753), which, while marginally lower than the more complex MELD score, still reflects moderate to good discriminative performance. These findings suggest that DBAR serves as a reliable prognostic tool, exhibiting predictive accuracy comparable to the established and widely validated MELD score within the same patient population. Furthermore, DBAR exhibited consistent predictive capacity for

**Table 3. Association between DBAR and mortality in patients with cirrhosis.**

| Variables | Model 1 | | | Model 2 | | | Model 3 | | |
|---|---|---|---|---|---|---|---|---|---|
| | HR | 95% CI | p-value | HR | 95% CI | p-value | HR | 95% CI | p-value |
| DBAR | 1.21 | 1.15, 1.26 | <0.001 | 1.15 | 1.09, 1.21 | <0.001 | 1.17 | 1.09, 1.26 | <0.001 |
| DBAR Group | | | | | | | | | |
| DBAR<4 | 1(Reference) | | 1(Reference) | 1(Reference) | | 1(Reference) | 1(Reference) | | 1(Reference) |
| DBAR≥4 | 4.82 | 3.22, 7.23 | <0.001 | 3.32 | 2.10, 5.24 | <0.001 | 3.05 | 1.87, 4.97 | <0.001 |

Model 1: adjusted for age, gender, and race;

Model 2: adjusted for age, gender, race, BUN, creatinine, and SIRS;

Model 3: adjusted for age, gender, race, BUN, creatinine, SIRS, potassium, sodium, lactate, WBC, platelet, INR, Sepsis, AKI, SBP, HRS, Variceal bleeding, Ascites, Vasoactive, MV, and CRRT.

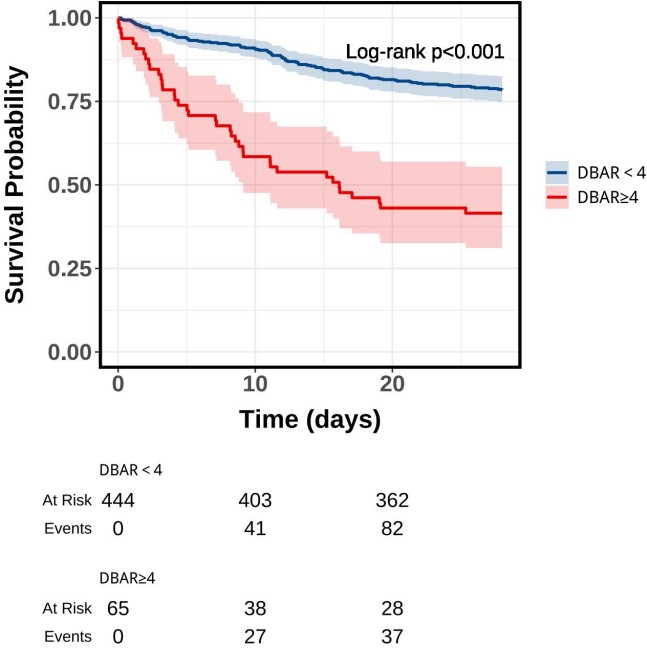

**Fig 2. Kaplan-Meier survival analysis curves for all-cause mortality in patients with cirrhosis at 28-d of hospital admission.**

in-hospital mortality (AUC = 0.720) and 90-day mortality (AUC = 0.676). Comprehensive data are presented in Fig 3 and Table 4.

### 3.5 DBAR feature importance in cirrhosis prognosis (Boruta analysis)

To comprehensively assess the significance of variables in predicting cirrhosis prognosis, we employed Boruta Feature Selection, a widely recognized machine learning algorithm, to identify key predictors. The analysis highlighted lactate, blood urea nitrogen, and DBAR as the top three most influential factors. This result further substantiates the robust relationship between DBAR and cirrhosis prognosis. A detailed representation of these findings is provided in Fig 4.

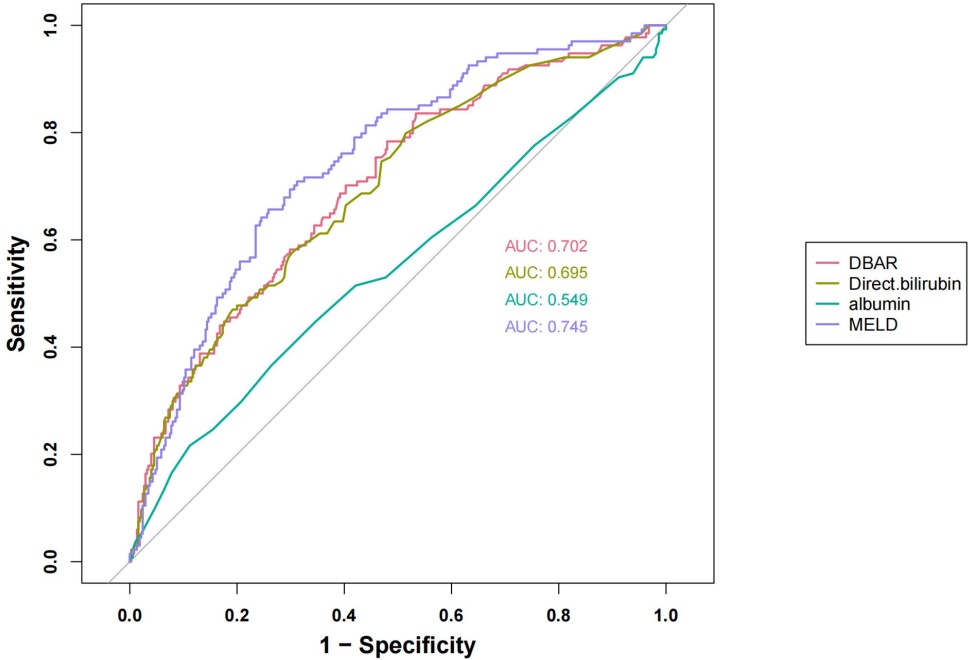

**Fig 3. DBAR and 28-day all-cause mortality: ROC curves.**

**Table 4. Details of ROC curves shown in Fig 3.**

| Variable | AUC | 95%CI | Threshold | Sensitivity | Specificity |
|---|---|---|---|---|---|
| DBAR | 0.702 | 0.650 - 0.753 | 0.777 | 0.784 | 0.520 |
| Direct bilirubin | 0.695 | 0.643 - 0.747 | 2.000 | 0.799 | 0.485 |
| Albumin | 0.549 | 0.489 - 0.609 | 2.200 | 0.216 | 0.888 |
| MELD | 0.744 | 0.697 - 0.792 | 22.619 | 0.709 | 0.689 |

### 3.6 RCS analysis of DBAR in relation to cirrhosis prognosis

Our study utilized RCS analysis to uncover a notable nonlinear association between the DBAR and cirrhosis prognosis, with P-values for nonlinearity and overall significance of 0.022 and <0.001, respectively. This finding indicates that higher DBAR levels are significantly associated with an increased risk of adverse clinical outcomes. For a comprehensive visualization of these results, please refer to Fig 5.

### 3.7 Clinical outcomes by DBAR subgroup in cirrhosis

In the subgroup analyses, the influence of demographic and clinical variables—including age, sex, race and sepsis—on patient outcomes was evaluated. No statistically significant interactions between these variables and the DBAR were observed. However, the predictive performance of the DBAR exhibited variability across subgroups defined by the presence of ascites and hepatorenal syndrome, indicating that its prognostic utility may be contingent upon these specific clinical conditions. Notably, the presence of ascites emerged as a potential explanatory factor for the differential predictive accuracy observed. The comprehensive results of these analyses are presented in Fig 6.

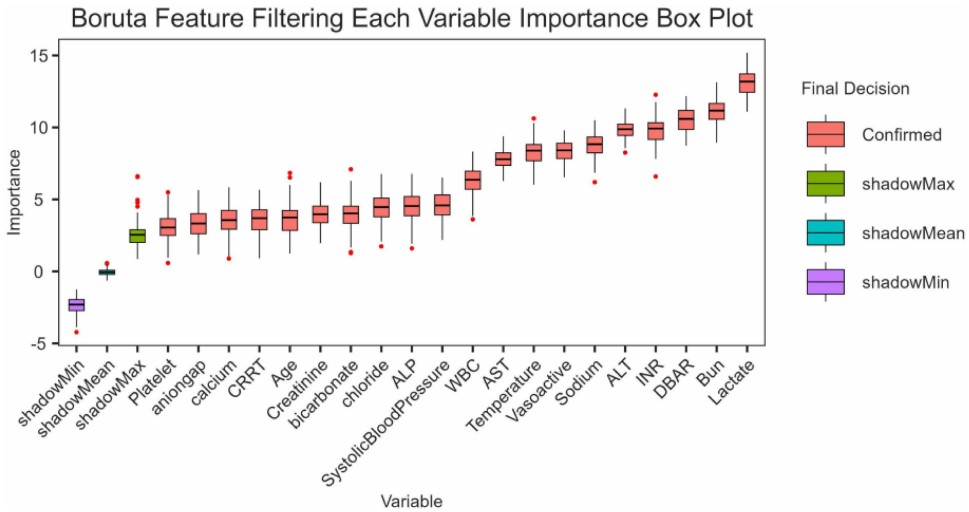

**Fig 4. DBAR and cirrhosis prognosis: Boruta feature importance analysis.**

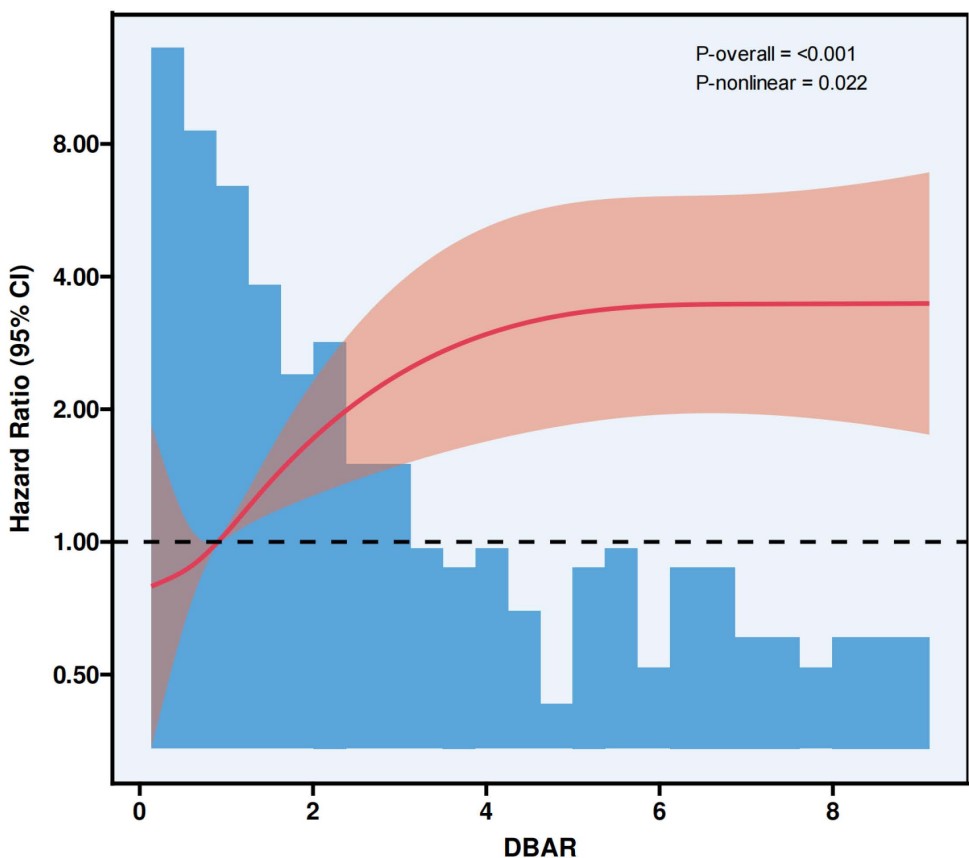

**Fig 5. DBAR and survival: RCS analysis.**

| Subgroup | DBAR < 4 | DBAR≥4 | | Adjusted HR (95% CI)* | P value | P for interaction |
|---|---|---|---|---|---|---|
| Overall | 96/444 (21.6) | 38/65 (58.5) | | 3.05 (1.87, 4.97) | <0.001 | |
| Age | | | | | | 0.285 |
| <60 | 40/224 (17.9) | 27/48 (56.2) | | 2.89 (1.54, 5.43) | 0.001 | |
| ≥60 | 56/220 (25.5) | 11/17 (64.7) | | 3.55 (1.66, 7.59) | 0.001 | |
| Gender | | | | | | 0.731 |
| Female | 35/133 (26.3) | 13/21 (61.9) | | 3.63 (1.57, 8.39) | 0.003 | |
| Male | 61/311 (19.6) | 25/44 (56.8) | | 2.86 (1.52, 5.39) | 0.001 | |
| Race | | | | | | 0.085 |
| OTHER | 37/146 (25.3) | 17/29 (58.6) | | 2.23 (1.00, 4.96) | 0.049 | |
| WHITE | 59/298 (19.8) | 21/36 (58.3) | | 3.64 (1.85, 7.17) | <0.001 | |
| Sepsis | | | | | | 0.661 |
| No | 16/90 (17.8) | 3/8 (37.5) | | 2.54 (0.35, 18.63) | 0.359 | |
| Yes | 80/354 (22.6) | 35/57 (61.4) | | 3.25 (1.93, 5.47) | <0.001 | |
| Ascites | | | | | | <0.001 |
| No | 28/203 (13.8) | 13/17 (76.5) | | 22.79 (6.88, 75.51) | <0.001 | |
| Yes | 68/241 (28.2) | 25/48 (52.1) | | 1.81 (0.98, 3.34) | 0.057 | |
| Hepatorenal syndrome | | | | | | 0.003 |
| No | 72/383 (18.8) | 27/46 (58.7) | | 4.20 (2.42, 7.31) | <0.001 | |
| Yes | 24/61 (39.3) | 11/19 (57.9) | | 1.75 (0.54, 5.71) | 0.352 | |
| Vasoactive | | | | | | 0.735 |
| No | 27/242 (11.2) | 11/27 (40.7) | | 4.05 (1.40, 11.70) | 0.01 | |
| Yes | 69/202 (34.2) | 27/38 (71.1) | | 2.90 (1.62, 5.19) | <0.001 | |

0.4  1  2.7  7.4  20.1  54.6

** adjusted for Age, Gender, Race, Bun, Potassium, Sodium, Creatinine, Lactate, WBC, Platelet, INR, AKI, Sepsis, Spontaneous bacterial peritonitis, Hepatorenal syndrome, SIRS, Variceal bleeding, Ascites, Vasoactive, Ventilator,

**Fig 6. Forest plots showing subgroup effects of DBAR on 28-day all-cause mortality in cirrhosis.**

### 3.8 External validation using eICU-CRD cohort

To validate the robustness of our findings, we further analyzed an independent validation cohort from the eICU Collaborative Research Database (eICU-CRD), a large multicenter critical care database comprising over 200,000 ICU admissions across the United States. Consistent with the results from the MIMIC-IV cohort, DBAR remained an independent predictor of short-term mortality in the eICU-CRD population after adjustment in multivariable Cox regression models. RCS analysis confirmed a nonlinear dose–response association between DBAR and short-term mortality risk. Moreover, ROC curve analysis demonstrated good discriminative ability, with an AUC of approximately 0.71. These findings further corroborate the robustness and generalizability of DBAR as a prognostic biomarker in critically ill cirrhotic patients.

Importantly, the prognostic value of DBAR was further confirmed in an independent external validation cohort (eICU-CRD), underscoring its robustness and generalizability across diverse ICU populations.

The restricted cubic spline analysis and ROC analysis of the validation cohort are provided as S1 and S2 Figs. The detailed results of the multivariable Cox regression models are presented in S1 Table.

## 4 Discussion

Cirrhosis, a chronic liver disease with significant global prevalence, is closely linked to elevated mortality rates, especially in individuals with cirrhosis [19]. Within ICU settings, prognostic scoring systems such as the MELD and the Child-Pugh score are standard tools for assessing the prognosis of this population. Emerging research has identified several specific ratios—including the international normalized ratio to albumin ratio [20], lactate to albumin ratio [21], and neutrophil to

albumin ratio [22]—as independent risk factors for cirrhosis, thereby improving prognostic accuracy. Nevertheless, there remains a pressing clinical need for prognostic indicators that are both simplified and more easily interpretable to enhance practical utility.

In this study, univariate and multivariate prognostic analyses were performed on a cohort of critically ill cirrhotic patients, revealing that the DBAR serves as an independent predictor of short-term all-cause mortality. Patients were categorized into high-risk (DBAR ≥4) and low-risk (DBAR <4) groups based on DBAR thresholds. KM survival analysis demonstrated a persistently significant prognostic disparity in the high-risk group, even after adjusting for multiple covariates (HR: 3.05, p < 0.001). Both DBAR and the MELD score exhibited moderate discriminative capacity in predicting 28-day mortality, with consistent predictive accuracy for in-hospital (ROC: 0.720) and 90-day mortality (ROC: 0.676). These findings underscore the potential clinical significance of DBAR in evaluating both short- and medium-to-long-term prognoses in patients with advanced cirrhosis admitted to the ICU. RCS analysis revealed a significant nonlinear association between DBAR and cirrhosis. Hazard ratios increased progressively with higher DBAR values. Subgroup analysis revealed variability in the predictive efficacy of DBAR among patients with ascites and HRS. This variability can be attributed to two primary factors: First, severe ascites is frequently associated with renal sodium retention and diminished effective circulating blood volume, which may exacerbate renal dysfunction, disrupt albumin metabolism, and impair bilirubin clearance. Second, the imperative for aggressive interventions, such as paracentesis and albumin supplementation, in these patients may introduce interaction effects that influence prognosis. Importantly, the prognostic significance of DBAR was further confirmed in an independent external validation cohort (eICU-CRD), which underscores the robustness and generalizability of our findings across diverse ICU populations. This study elucidates the potential role of DBAR as a prognostic biomarker in critically ill cirrhotic patients, while highlighting the complex interplay of pathophysiological mechanisms in advanced liver disease.

The Albumin-Bilirubin (ALBI) score, primarily utilized for grading liver cancer prognosis, has also found application in assessing critically ill cirrhotic patients [14]. In a case-control study involving 204 patients with liver failure, Li et al. [16] demonstrated that the ratio of indirect bilirubin to albumin is significantly associated with the prognosis of hepatic encephalopathy (HR: 1.626, P < 0.001), underscoring the prognostic relevance of the bilirubin-to-albumin ratio in cirrhosis-related complications. While direct bilirubin, a marker of hepatic metabolic function, has been strongly linked to cirrhosis prognosis, indirect bilirubin—often elevated due to hemolytic reactions—plays a less critical role in cirrhotic patient survival compared to intrahepatic cholestasis and impaired hepatic clearance [17]. This suggests that direct bilirubin levels are more strongly predictive of poor prognosis in cirrhosis than indirect bilirubin (OR:1.373). Furthermore, a separate study highlighted the prognostic utility of the direct bilirubin to total bilirubin ratio in liver failure [23].

Serum albumin, exclusively synthesized by the liver, serves as a key indicator of hepatic synthetic function and nutritional status [12]. Hypoalbuminemia, commonly observed in cirrhotic patients [24,25], is strongly associated with increased mortality [3] and can exacerbate complications such as ascites and edema, thereby worsening the clinical course of the disease [26,27]. This condition may arise from inflammation-induced extravascular albumin loss, impaired hepatic synthesis, or malnutrition, all of which heighten infection risk and may signal impending organ failure [28,29]. Low albumin levels have also been linked to poor outcomes in acutely ill patients [30,31], further emphasizing its prognostic significance. Consequently, hypoalbuminemia is closely correlated with disease severity and adverse prognosis in cirrhosis. To leverage the complementary prognostic value of bilirubin and albumin, we utilized an inverse variation mechanism to mitigate the impact of individual indicators, specifically through the direct bilirubin to albumin ratio.

The complexity of certain ICU scoring systems can hinder their practical application in the routine management of critically ill cirrhotic patients. Widely used scoring systems, such as MELD and Child-Pugh, may be compromised by the subjectivity associated with evaluating hepatic encephalopathy. In contrast, the DBAR offers simplicity without compromising accuracy. Our study demonstrates that elevated DBAR levels are strongly associated with the severity of cirrhosis and

poor prognosis, establishing it as a reliable indicator for assessing patient condition and outcomes. By integrating information on liver function, inflammation, and nutritional status, DBAR provides more comprehensive insights compared to isolated evaluations of direct bilirubin or albumin. An elevated DBAR reflects increased direct bilirubin levels and decreased albumin levels—two critical biomarkers. Elevated direct bilirubin suggests impaired hepatic function and disease severity, while reduced albumin levels may indicate impaired liver synthesis, albumin loss, inflammatory consumption, or malnutrition, all of which are linked to poor prognosis. DBAR effectively leverages the combined prognostic value of direct bilirubin and albumin in severe cirrhosis. Clinicians can use DBAR values to guide treatment decisions, employing standard therapy for patients below the threshold and opting for more intensive interventions for those above it. This stratification enables the efficient identification of severe cases and optimizes the allocation of medical resources, thereby improving the precision and effectiveness of clinical care. Furthermore, since direct bilirubin and albumin are routinely measured in standard blood tests, calculating DBAR does not require additional laboratory work, minimizing the financial burden on patients. Overall, DBAR emerges as a valuable prognostic tool for assessing severe cirrhosis by synergistically combining direct bilirubin and albumin levels. Its ease of use, cost-effectiveness, and robust predictive capacity make it a practical and reliable indicator in clinical practice.

In this study, we utilized data from publicly available databases to investigate prognostic factors associated with the disease. The primary strength of this approach is the access to a comprehensive real-world dataset; however, several significant limitations must be addressed. First, the inclusion of patients without severity stratification restricts our capacity to evaluate potential differences across populations with varying disease severity. Second, the absence of detailed etiology data for cirrhosis in critically ill cirrhotic patients further impedes our analysis of variations among patient groups with different underlying causes. Additionally, we were unable to determine the precise cause of death for all patients; thus, the observed mortality rate in critically ill patient

individuals may be influenced by multiple factors, and our analysis treats this as all-cause mortality. Furthermore, the study population exclusively comprised patients admitted to the ICU, a characteristic that may introduce selection bias and limit the generalizability of our conclusions.

## 5 Conclusion

DBAR emerges as an independent prognostic marker for mortality among critically ill cirrhotic patients and represents a readily accessible laboratory parameter for identifying this high-risk population. However, additional large-scale, prospective, multicenter studies are warranted to comprehensively evaluate and validate its clinical applicability.

## Supporting information

**S1 File. The compiled database used for all analyses in this study.**
(RAR)

**S1 Table. Association between DBAR and mortality in the eICU-CRD validation cohort.**
(DOCX)

**S1 Fig. DBAR and mortality in the eICU-CRD cohort (RCS analysis).**
(TIF)

**S2 Fig. DBAR and mortality in the eICU-CRD cohort (ROC analysis).**
(TIF)

## Acknowledgments

Not applicable.

 

## Author contributions

**Data curation:** XingYi Yang, Ji Yang.

**Formal analysis:** XingYi Yang, Ji Yang, GuangDong Wang.

**Software:** Ji Yang, LiHong Lv.

**Writing – original draft:** XingYi Yang, Zhang Min.

**Writing – review & editing:** XingYi Yang, Ji Yang.

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
